# Novel Peritoneal Sclerosis Rat Model Developed by Administration of Bleomycin and Lansoprazole

**DOI:** 10.3390/ijms242216108

**Published:** 2023-11-09

**Authors:** Kosei Kunitatsu, Yuta Yamamoto, Shota Nasu, Akira Taniji, Shuji Kawashima, Naoko Yamagishi, Takao Ito, Shigeaki Inoue, Yoshimitsu Kanai

**Affiliations:** 1Department of Emergency and Critical Care Medicine, Wakayama Medical University, 811-1 Kimiidera, Wakayama 641-8509, Japan; 2Department of Anatomy and Cell Biology, Wakayama Medical University, 811-1 Kimiidera, Wakayama 641-8509, Japan

**Keywords:** fibrosis, collagen 1a1, bleomycin, lansoprazole, *Mcp1*, *Mcp3*, *Opn*

## Abstract

In our preliminary experiment, peritoneal sclerosis likely induced by peritoneal dialysis was unexpectedly observed in the livers of rats given bleomycin and lansoprazole. We examined whether this peritoneal thickening around the liver was time-dependently induced by administration of both drugs. Male Wistar rats were injected with bleomycin and/or lansoprazole for 2 or 4 weeks. The 3YB-1 cell line derived from rat fibroblasts was treated by bleomycin and/or lansoprazole for 24 h. The administration of both drugs together, but not individually, thickened the peritoneal tissue around the liver. There was accumulation of collagen fibers, macrophages, and eosinophils under mesothelial cells. Expressions of *Col1a1*, *Mcp1* and *Mcp3* genes were increased in the peritoneal tissue around the liver and in 3YB-1 cells by the administration of both drugs together, and *Opn* genes had increased expressions in this tissue and 3YB-1 cells. Mesothelial cells indicated immunoreactivity against both cytokeratin, a mesothelial cell marker, and αSMA, a fibroblast marker, around the livers of rats given both drugs. Administration of both drugs induced the migration of macrophages and eosinophils and induced fibrosis associated with the possible activation of fibroblasts and the possible promotion of the mesothelial–mesenchymal transition. This might become a novel model of peritoneal sclerosis for peritoneal dialysis.

## 1. Introduction

The peritoneum is the largest serous membrane; it partially or fully covers the intra-abdominal organs [1]. It is contained within an epithelial monolayer of mesothelial cells and loose connective tissues including fibroblasts [2]. Peritoneal dialysis is a renal replacement therapy for renal failure; it slowly induces inflammation and fibrosis in the peritoneum [3,4]. Sub-mesothelial connective tissue is increased in thickness by peritoneal dialysis in a time-dependent manner [5]. In rat models of peritoneal dialysis using microbicides and alcohol, there was reportedly an increased expression of monocyte chemoattractant protein (Mcp)-1 and transforming growth factor (Tgf)-β1. This increase was inhibited by a specific compound which ameliorated peritoneal thickening [6]. Peritoneal fibrosis induced by peritoneal dialysis in the long term may therefore be associated with the migration of macrophages and the production of the extracellular matrix, including type I collagen.

Lansoprazole is a proton pump inhibitor used for gastric and esophageal ulcers [7]. The expression of anti-oxidative stress proteins is increased in response to reactive oxygen species (ROS) via the nuclear factor erythroid 2-related factor 2 (Nrf2) pathway, and the upregulation of these genes ameliorates the cell damage induced by ROS. Lansoprazole induction was also reported to increase the expression of antioxidative stress protein genes through the activation of Nrf2 [8]. Lansoprazole ameliorated inflammation in the intestines, liver, and kidneys via the upregulation of anti-oxidative stress protein genes [8,9,10]. Lansoprazole inhibited the increased upregulation of interleukin 6 in rat hearts injured by cisplatin and fibrosis in non-alcoholic steatohepatitis model rats [11,12]. Bleomycin is an anticancer drug for squamous cell carcinoma, and it is used as an ROS inducer or for making an animal model of pulmonary inflammation [13]. We therefore examined whether lansoprazole ameliorates the pulmonary inflammation induced by bleomycin. Our preliminary experiment incidentally found that the peritoneal tissue was remarkably thick, a gross abnormality, around the livers of rats given both bleomycin and lansoprazole (BLM + LAP) subcutaneously for 4 weeks.

This study examined whether the peritoneal thickening around the liver was time-dependently induced by administration of BLM + LAP. To understand the mechanism, we analyzed histological changes in the peritoneal tissue around rat livers. To explore the genes associated with these histological changes, changes in gene expression induced by BLM + LAP treatment were analyzed in the livers of rats and 3Y1-B cells derived from rat fibroblast cells.

## 2. Results

### 2.1. Peritoneal Changes around the Livers of Rats Given BLM + LAP

#### 2.1.1. Histological Changes in the Peritoneal Tissue around the Liver

We examined whether tissue injury induced by bleomycin was ameliorated by lansoprazole, which has been shown to have anti-inflammatory effects in several organs [8,9,10,12]. The peritoneal tissue at the inferior border of the liver thickened in a time-dependent manner (Figure 1A). We analyzed the liver histologically with hematoxylin eosin and Masson–Goldner staining. Reactive mesothelial cells were observed in rats given BLM + LAP for 2 weeks (Figure 1B), and collagen fiber under the mesothelial cells was observed in two out of the six rats given BLM + LAP for 2 weeks and in all rats given BLM + LAP for 4 weeks. Collagen fiber under the mesothelial cells of peritoneal tissue around the liver was not observed in rats given either bleomycin or lansoprazole separately (Figure 1E). The thickness of peritoneal tissue around the liver increased in rats given BLM + LAP in a time-dependent manner (Figure 1C). A two-way analysis of variance (ANOVA) indicated that the effects of lansoprazole (2 weeks: F(1, 17) = 0.00, *p* = 0.96, 4 weeks: F(1, 17) = 0.06, *p* = 0.81) and bleomycin (2 weeks: F(1, 17) = 0.00, *p* = 0.98, 4 weeks: F(1, 17) = 0.01, *p* = 0.91) were not significant, but the effect of their interaction was significant (2 weeks: F(1, 17) = 5.83, *p* < 0.05, 4 weeks: F(1, 17) = 28.13, *p* < 0.05). The migration of eosinophils (Figure 1B: indicated by arrowheads) and macrophages (Figure 1D) was detected in the thickened peritoneal tissue of rats given BLM + LAP for 4 weeks.

#### 2.1.2. Effect of BLM + LAP on Rat Liver Injuries

Serum AST and ALT concentrations were measured in rats administered these drugs for 4 weeks to evaluate their effect on liver injuries (Figure 2A). A two-way ANOVA indicated no significant effects of lansoprazole (F(1, 17) = 0.05, *p* = 0.82) or bleomycin (F(1, 17) = 2.42, *p* = 0.14) and their interaction (F(1, 17) = 3.30, *p* = 0.09) in AST. There were no significant effects of lansoprazole (F(1, 17) = 4.09, *p* = 0.06) or bleomycin (F(1, 17) = 0.02, *p* = 0.89), but a significant difference in their interaction (F(1, 17) = 6.62, *p* < 0.05) was revealed in ALT. Dunnet’s test indicated that there were no significant differences in the concentrations of AST and ALT compared with the control group. The expressions of *HO-1*, *catalase (Cat)*, *glutathione S-transferase alpha 2* (*Gsta2*), *NAD(P)H quinone dehydrogenase 1* (*Nqo1*), and *glutathione peroxidase 1* (*Gpx1*) genes were also measured via quantitative RT-PCR. The expression of the *Nqo1* gene was significantly increased in the lansoprazole, bleomycin and BLM + LAP groups (Appendix A). To evaluate the effect of the secretion of bile due to peritoneal thickening, we measured the total bilirubin (Figure 2B). A two-way ANOVA indicated that there were no significant effects of lansoprazole (F(1, 17) = 1.73, *p* = 0.21), bleomycin (F(1, 17) = 0.14, *p* = 0.71), or their interaction (F(1, 17) = 2.17, *p* = 0.16) in total bilirubin. Dunnet’s test indicated that there were no significant differences in the concentrations compared with the control group. There was no detection of γGTP in the serum of any of the rats.

#### 2.1.3. Expression Changes of Genes Associated with Fibrosis in the Peritoneal Tissue around the Liver

To explore the mechanism of peritoneal tissue thickening, we measured the expression of *collagen type 1 alpha 1 chain* (*Col1a1*) and *transforming growth factor beta 1* (*Tgfb1*) genes. A two-way ANOVA indicated significant differences in the effect of drug interactions on the expression of *Col1a1* genes in the liver of rats administered drugs for 4 weeks (lansoprazole (2 weeks: F(1, 16) = 2.91, *p* = 0.11, 4 weeks: F(1, 17) = 0.05, *p* = 0.83), bleomycin (2 weeks: F(1, 16) = 3.58, *p* = 0.08, 4 weeks: F(1, 17) = 0.45, *p* = 0.51) and their interaction (2 weeks: F(1, 16) = 2.10, *p* = 0.17, 4 weeks: F(1, 17) = 8.01, *p* < 0.05)). Dunnett’s test indicated that the expression of *Col1a1* was significantly increased only in the livers of rats given BLM + LAP for 4 weeks (Figure 3A). A two-way ANOVA indicated significant differences in the effect of lansoprazole on the expression of *Tgfb1* genes in the liver of rats administered drugs for 2 weeks (lansoprazole (2 weeks: F(1, 16) = 5.91, *p* < 0.05, 4 weeks: F(1, 17) = 1.84, *p* = 0.19), bleomycin (2 weeks: F(1, 16) = 0.20, *p* = 0.66, 4 weeks: F(1, 17) = 2.57, *p* = 0.13) and their interaction (2 weeks: F(1, 16) = 4.01, *p* = 0.06, 4 weeks: F(1, 17) = 0.65, *p* = 0.43)). Dunnett’s test indicated that there was no change in expression of *Tgfb1* in any of the groups (Figure 3B). To examine whether these changes in expression occurred in the liver ubiquitously, we measured the expression of these genes in the outer and inner parts of liver (Figure 3C). A two-way ANOVA indicated significant differences in the effects of drug interaction on the expression of *Col1a1* genes in the outer parts of liver in rats administered drugs for 4 weeks (lansoprazole (F(1, 17) = 0.01, *p* = 0.94), bleomycin (F(1, 17) = 0.26, *p* = 0.62) and their interaction (F(1, 17) = 7.65, *p* < 0.05)). However, there were no significant differences in any effects of the drug or their interaction on the expression of *Col1a1* genes in the inner parts of livers from rats treated for 4 weeks (lansoprazole (F(1, 17) = 2.27, *p* = 0.15), bleomycin (F(1, 17) = 0.41, *p* = 0.53) and their interaction (F(1, 17) = 0.00, *p* = 0.96)). Dunnett’s test indicated that an increase in *Col1a1* gene expression was observed in the outer parts of the livers of rats that were given BLM + LAP for 4 weeks, but it was not observed in the inner parts of the livers (Figure 3D).

#### 2.1.4. Expression Changes of Genes Associated with Migration of Macrophages and Eosinophils in the Peritoneal Tissue around the Liver

Migration of macrophages and eosinophils was observed in the peritoneal tissue thickened by BLM + LAP. Ip10, Mip1a1, Mcp1, Mcp3 and Rantes promoted the migration of macrophages, and Mcp3 and Rantes also promoted the migration of eosinophils. We measured the expression of chemokine genes expressed in fibroblasts. Focusing on the outer part, a two-way ANOVA indicated that a synergy between lansoprazole and bleomycin was observed in the upregulation of *Mcp1* genes (lansoprazole (F(1, 16) = 0.00, *p* = 0.98), bleomycin (F(1, 16) = 0.63, *p* = 0.44) and their interaction (F(1, 16) = 9.47, *p* < 0.05)) but not *Mcp3* genes (lansoprazole (F(1, 16) = 0.00, *p* = 0.98), bleomycin (F(1, 16) = 1.41, *p* = 0.25) and their interaction (F(1, 12) = 2.52, *p* = 0.13)) in the outer part of the livers from rats treated for 4 weeks (Appendix A). Dunnett’s test indicated that the expressions of *Mcp1* and *Mcp3* genes tended to be increased by BLM + LAP after 2 weeks (Figure 4A), and they were significantly increased after 4 weeks (Figure 4B). However, upregulation of *Mcp1* and *Mcp3* genes was not observed in the inner part (Figure 4C,D). Expression of *Mcp1* and *Mcp3* genes was also significantly increased in 3Y1-B cells derived from rat fibroblast cells treated by BLM + LAP (Figure 4E), although synergy between lansoprazole and bleomycin was observed in the upregulation of *Mcp1* genes (lansoprazole (F(1, 12) = 3.38, *p* = 0.09), bleomycin (F(1, 12) = 9.23, *p* < 0.05) and their interaction (F(1, 12) = 6.42, *p* < 0.05)) but not *Mcp3* genes (lansoprazole (F(1, 12) = 2.50, *p* = 0.14), bleomycin (F(1, 12) = 1.85, *p* = 0.20) and their interaction (F(1, 12) = 2.17, *p* = 0.17)).

### 2.2. BLM + LAP Increased the Expression of Genes Associated with the Migration and Proliferation of Fibroblasts

Activation of fibroblasts might be promoted in the peritoneal tissues of the liver in the BLM + LAP group because collagen fiber was accumulated in the peritoneal membrane (Figure 1D). The osteopontin (*Opn*) gene is expressed in the activated fibroblasts. A two-way ANOVA indicated significant differences in the effect of drug interactions on the expression of the *Opn* gene in the outer parts of the livers of rats treated for 4 weeks (lansoprazole (F(1, 16) = 0.36, *p* = 0.55), bleomycin (F(1, 16) = 0.23, *p* = 0.64) and their interaction (F(1, 16) = 7.27, *p* < 0.05)). Meanwhile, there were significant differences in the effect of lansoprazole on the expression of *Opn* in 3YB-1 cells treated for 24 h (lansoprazole (F(1, 12) = 6.82, *p* < 0.05), bleomycin (F(1, 12) = 2.35, *p* = 0.15) and their interaction (F(1, 12) = 0.34, *p* = 0.57)). Expression of the *Opn* gene was significantly increased in the outer part of the livers of rats that were given BLM + LAP for 4 weeks (Figure 5A). This increase was also detected in 3Y1-B cells (Figure 5B).

### 2.3. BLM + LAP Increased the Number of Mesothelial Cells Indicating Immunoreactivity against Cytokeratin and αSMA.

Reactive mesothelial cells were detected in the peritoneal tissue of the livers in the BLM + LAP group after 2 weeks (Figure 1B). To examine whether the reactive mesothelial cells were differentiated to fibroblasts, immunohistochemistry was performed in the liver tissue of rats administered BLM + LAP for 2 weeks with anti-cytokeratin, which is expressed in mesothelial cells and anti-α smooth muscle actin (αSMA), which is expressed in fibroblasts. The form of mesothelium cells, excluding one case in the bleomycin group, was not changed with lansoprazole or bleomycin, and reactive mesothelial cells indicated immunoreactivity against both cytokeratin and αSMA in the BLM + LAP group (Figure 6) and one of the bleomycin groups (Appendix A). The immunoreactivity against cytokeratin was weak in one case in which the peritoneal tissue was thickened (Appendix A).

## 3. Discussion

This study demonstrated that the administration of BLM + LAP time-dependently thickened the peritoneal tissue around rat livers. Thickening of the peritoneal tissue was induced by the infiltration of macrophages and eosinophils and the accumulation of collagen fibers (Figure 1). This phenotype might be associated with the upregulation of *Col1a1*, *Mcp1* and *Mcp3* genes in the peritoneal tissue of the liver (Figure 3 and Figure 4). The expression of the *Opn* gene, which is expressed in activated fibroblasts, was increased by BLM + LAP in 3YB-1 cells derived from rat fibroblast cells, as well as in peritoneal liver tissues (Figure 5). Immunohistochemistry indicated the possibility that the reactive mesothelial cells might also induce the mesothelial–mesenchymal transition (MMT) by BLM + LAP (Figure 6). Administration of BLM + LAP could therefore affect the fibroblasts in the peritoneal tissue regarding gene expression changes associated with peritoneal thickening, and it might be associated with a MMT in mesothelial cells (Figure 7).

Bleomycin, the agonist of Toll-like receptor (TLR) 2, induced inflammation and fibrosis, as shown by the expression of proinflammatory cytokines and Tgfb1 in mouse lungs [13,14]. This fibrosis resulted in the activation of fibroblasts [15,16]. However, the reported bleomycin-induced fibrosis was pulmonary fibrosis, not pleural fibrosis. In this study, peritoneal fibrosis in the liver, but not liver fibrosis, was revealed to be time dependent in some of the rats given BLM + LAP (Figure 1). The phenotype was observed in the rats given either bleomycin or lansoprazole for 28 days (Figure 1). This phenotype may therefore be different from the previously reported bleomycin-induced fibrosis. Peritoneal fibrosis induced by the administration of BLM + LAP for 28 days might induce bile statis, so we measured the concentrations of γGTP and bilirubin in serum. The concentration of γGTP was lower than the detection limit in all rats, and the concentration of bilirubin was not increased by the administration of BLM + LAP (Figure 2). Peritoneal fibrosis may therefore be unnoticeable if the fibrosis occurs in patients undergoing chemotherapy with bleomycin and taking lansoprazole for gastric ulcers.

Administration of BLM + LAP increased the expression of chemokine genes (*Mcp1* and *Mcp3*), which promote the migration of macrophages and eosinophils and promote the activation of fibroblasts, which expressed the *Opn* gene in sub-mesothelial connective tissues. The mesothelial–mesenchymal transition might be progressed by the administration of BLM + LAP.

Bleomycin as an ROS inducer might injure the hepatocytes in the liver, but no effect of bleomycin and/or lansoprazole on the toxicity of hepatocytes was observed in serum AST and ALT levels (Figure 2A). The expression of the *Nqo1* gene was increased by lansoprazole and bleomycin (Appendix A). The upregulation of the *Nqo1* gene might be induced by lansoprazole via activation of Nrf2 or ROS induced by bleomycin. Thus, there might be no heavy cellular damage in the livers of rats given bleomycin due to the cytoprotection effect of Nqo1. The peritoneal sclerosis in this study induced by BLM + LAP might not be associated with the remarkable exposure to ROS.

Peritoneal dialysis over a long time induced encapsulating peritoneal sclerosis (EPS), the accumulation of collagen fiber, and the migration of macrophages and eosinophils [17,18]. The expression of Col1a1 was increased in patients that underwent peritoneal dialysis [19]. The peritoneal sclerosis induced by the administration of BLM + LAP had histological changes similar to that of EPS (Figure 1). The expressions of *Col1a1*, *Mcp1* and *Mcp3* genes in peritoneal tissues of the liver were increased by the administration of BLM + LAP (Figure 3 and Figure 4). The migration of macrophages is associated with several chemokines. The chemokines which were expressed in fibroblasts and induced the migration of macrophages were IP-10 [20,21], Mcp1 [22,23], Mcp3 [24,25] and Rantes [24]. Eosinophils are also attracted by Mcp3 and Rantes [26]. The accumulation of collagen fiber might therefore be associated with the upregulation of the *Col1a1* gene, and the migration of macrophages might be associated with the upregulation of the *Mcp1* and *Mcp3* genes. The migration of eosinophils might also be associated with the upregulation of the *Mcp3* gene. Expression changes of these genes were observed in 3YB-1 cells treated with BLM + LAP. BLM + LAP might therefore increase the expression of Mcp1 and Mcp3 genes in the fibroblast cells of the peritoneum following the migration of macrophages and eosinophils.

Opn is a cytokine which activates fibroblasts, producing extracellular matrix including collagen 1 [27,28]. Bleomycin induces the maturation of Tgfβ from latent Tgfβ in mouse lungs [29], and maturated Tgfβ induces the activation of fibroblasts which secrete Opn protein [30]. Extracellular matrix, including collagen type 1, is produced by activated fibroblasts [31]. Thus, the production of extracellular matrix is associated with the Tgfβ/SMAD and Tgfβ/non-SMAD pathways, including the Tgfβ/p38 pathway [32]. Bleomycin is also an agonist of TLR2 and promotes the production of inflammatory cytokines and chemokines [14,33]. Tgfβ/non-SMAD pathways including the Tgfβ/p38 pathway and the TLR2 pathway also include the TNF receptor-associated factor 6 (TRAF6). Lansoprazole enhances polyubiquitination of TRAF6 through binding lansoprazole to the deubiquitinate enzyme and cylindromatosis, and activates the Tgf-β-activated kinase-1 (TAK1)–p38 pathway [34]. Bleomycin might have promoted the production of Col1a1 via the non-SMAD pathway and the production of Mcp1 and Mcp3 via TLR2 signaling in the activated fibroblasts expressing Opn, but this phenomenon was not observed with the dosage and during the period we applied in this study (Figure 3, Figure 4 and Figure 5). Lansoprazole, as the activator of the TAK1-p38 pathway, might enhance the Tgfβ/p38 pathway in fibroblasts, and this might increase the expression of *Col1a1*, *Mcp1* and *Mcp3* in the liver around the peritoneum and fibroblast cells. An effect of drug interactions between lansoprazole and bleomycin on the upregulation of the *Mcp1* gene was observed in the outer part of the liver and fibroblast cells, but the effect of drug interactions between lansoprazole and bleomycin on the upregulation of the *Mcp3* gene was not observed. Thus, the upregulation of the *Mcp1* gene might be associated with the synergistic effect of BLM + LAP, while the upregulation of the *Mcp3* gene might be associated with the additive effect of BLM + LAP.

The MMT results from peritoneal dialysis and is induced by cytokines, including Tgfβ [35,36]. The expression of mesothelial cell marker proteins including cytokeratin is decreased and the expression of fibroblast marker proteins including αSMA is increased in MMT [35]. The mesothelial cells of hepatic peritoneal tissues indicated immunoreactivity against both cytokeratin and αSMA in rats given BLM + LAP for 14 days (Figure 6). Cells which were immunoreactive against both cytokeratin and αSMA were also detected in the hepatic peritoneal tissue in one out of five rats in the bleomycin group (Appendix A). Bleomycin induced the epithelial–mesenchymal transition via Tgfβ in mouse lungs [37]. Bleomycin might therefore slightly induce the MMT in the hepatic peritoneal tissue, but the MMT might be enhanced by lansoprazole via the Tgfβ/p38 pathway.

This study did not, however, examine whether administration of BLM + LAP promotes the MMT in mesothelial cells from peritoneal tissue, because mesothelial cell lines derived from peritoneal tissue are not deposited in cell banks. Future studies are required to demonstrate that the MMT is induced by bleomycin and enhanced by lansoprazole in mesothelial cells from peritoneal tissue. The peritoneal thickness might be enhanced by lansoprazole rather than other proton pump inhibitors because this enhancement is the result of the binding of lansoprazole to cylindromatosis [34]. Further study is required to examine whether the peritoneal sclerosis in the BLM + LAP group also occurs with the combination of bleomycin and other proton pump inhibitors. Peritoneal sclerosis might be associated with the activation and enhancement of the Tgfβ pathway, but there was no significant change in expression of the *Tgfb1* gene. One reason for this is that bleomycin might mature Tgfβ but not increase the expression of Tgfb1. Another reason could be that there was a comparatively small number of rats used to examine whether the expression of Tgfb1 was truly increased by the administration of BLM + LAP.

The animal model of peritoneal dialysis was prepared using peritoneal dialysates with high glucose and/or chlorhexidine gluconate levels [24,38]. These solutions indicate that the cytotoxicity, as well as bleomycin and high glucose solutions, may affect the blood glucose levels in animals. The administration of BLM + LAP does not directly affect the mesothelial cells in peritoneal tissue. This novel animal model of peritoneal sclerosis may aid in understanding the mechanism of the development of peritoneal fibrosis induced by peritoneal dialysis.

## 4. Materials and Methods

### 4.1. Chemicals

The following chemicals were purchased and used in this study: lansoprazole (129-05863, Fujifilm Wako, Osaka, Japan), bleomycin (21800AMX10210, Nippon Kayaku, Tokyo, Japan), and carboxymethyl cellulose (CMC) (039-01335, Fujifilm Wako).

### 4.2. Animals

Five-week-old male Wistar rats were purchased from Kiwa Laboratory Animals (Wakayama, Japan). Two or three rats were housed in a plastic rat cage (24.7 cm × 40.9 cm × 19.7 cm) with free access to tap water and laboratory animal feed (Oriental Yeast Co., Ltd., Tokyo, Japan) under a 12 h light/dark cycle (lights on/off at 8:00 a.m./p.m.) at 25 °C ± 1 °C and 50–60% humidity. All animals were used for experiments after an acclimation period of 1 week. 

### 4.3. Drug Administration

Twenty-one six-week-old male Wistar rats were divided into four groups (control (*n* = 5), lansoprazole (*n* = 5), bleomycin (*n* = 5), BLM + LAP (*n* = 6) groups) in every time course (2-week or 4-week experiment). Saline was subcutaneously injected into the right side of the back in control and lansoprazole groups, and bleomycin (1 mg/mL) was subcutaneously injected into the right side of the back in bleomycin and BLM + LAP groups. A 0.5% CMC solution was subcutaneously injected into the left side of the back in control and bleomycin groups, and a lansoprazole suspension with 0.5% CMC (30 mg/mL) was subcutaneously injected into the right side of the back in lansoprazole and BLM + LAP groups. Drug administration was performed over 14 or 28 days. One day after the last drug administration, blood samples were collected under anesthesia with isoflurane, and liver tissue samples were obtained following perfusion with 4% formalin neutral buffer solution.

### 4.4. Cell Culture

3Y1-B clone 1-6 (3Y1-B) cells were derived from a non-oncogenic rat fibroblast cell line (JCRB0734, Japanese Collection of Research Bioresources, Osaka, Japan) (PMID: 166944). Cells were maintained in Dulbecco’s modified Eagle medium (DMEM) without phenol red (Fujifilm Wako) and supplemented with 10% fetal bovine serum (Sigma-Aldrich, St. Louis, MO, USA) at 37 °C with 5% CO_2_. Lansoprazole was dissolved in dimethyl sulfoxide (DMSO) (Fujifilm Wako). Cells were treated for 24 h in the culture media with drugs (control media: 0.5% DMSO, media: 0.5% DMSO and 50 µM lansoprazole, media: 0.5% DMSO and 10 µg/mL bleomycin, and BLM + LAP media: 0.5% DMSO, 50 µM LAP and 10 µg/mL BLM). Cell culture experiments were performed four times.

### 4.5. Histology

Obtained liver tissues were fixed by 4% formalin neutral buffer solution for a day and embedded in paraffin. Specimens in paraffin blocks were cut at an interval of 5 µm and stained with hematoxylin and eosin solutions (Muto Pure Chemicals, Tokyo, Japan). Masson–Goldner staining was performed with an MG staining kit (Melck Millipore, Burlington, MA, USA).

For immunohistochemistry, the sections were incubated in HistoVT One (06380-05 Nacalai Tesque, Kyoto, Japan) for 20 min at 90 °C. The blocking process was performed with Blocking One (03953-66, Nacalai Tesque) after incubation in 0.3% H_2_O_2_/methanol for 30 min. Mouse anti-CD68 antibody (1:100, MCA341R, Bio-Rad Laboratories, Hercules, CA, USA) was used as the primary antibody, and anti-mouse IgG antibody (1:100, BA-2001, Vector Laboratories, Burlingame, CA, USA) was used as the secondary antibody. To detect the antibodies, we used an elite ABC kit (PK-6100, Vector Laboratories) and diaminobenzidine solutions (0.1 mg/mL, Dojindo Laboratories, Kumamoto, Japan).

For fluorescent immunohistochemistry, the sections were incubated in citric acid buffer (pH 6.0) for 15 min in a microwave. The blocking process was performed with 1% normal goat serum (S-1000, Vector Laboratories) in phosphate-buffered saline. Mouse anti-cytokeratin antibody (1:100, GTX75521, Gene Tex, Irvine, CA, USA) and rabbit anti-αSMA (1:400, GTX100034, Gene Tex) were used as the primary antibodies, and anti-mouse IgG antibody conjugated with Alexa Fluor (1:100, A-11029, Invitrogen, Waltham, MA, USA) and anti-rabbit IgG antibody conjugated with Alexa Fluor 578 (1:100, A-11011, Invitrogen) were used as the secondary antibody. To stain the nucleus, we used a 4′,6-diamidino-2-phenylindole (DAPI) solution (1:2000, Dojindo Laboratories).

### 4.6. Quantitative PCR

To extract RNA samples from tissue samples (approximately 20 mg), Sepasol RNA 1 Super G (Nacalai Tesque) was used. ReverTra Ace (FSQ-301, TOYOBO, Osaka, Japan) was used to synthesize first-strand cDNA samples from RNA samples (500 ng). Quantitative RT-PCR analyses were performed with the Brilliant III Ultra-Fast SYBR Green QPCR Master Mix (Agilent Technologies, Tokyo, Japan) and an AriaMX Real-time PCR system (Agilent Technologies). The primer sets used are indicated in Table 1. The relative expression ratio of each gene (gene/Actb) was normalized to that of the control group.

To extract RNA samples from cell samples, Sepasol RNA 1 Super G (Nacalai Tesque) was used. ReverTra Ace (FSQ-301, TOYOBO, Osaka, Japan) was used to synthesize first-strand cDNA samples from RNA samples (500 ng). Quantitative RT-PCR analyses were performed with a KAPA SYBR FAST qPCR Kit (KAPA Biosystems, Wilmington, MA, USA) and a CFX96 real-time PCR system (Bio-Rad Laboratories). The primer sets used are indicated in Table 1. The relative expression ratio of each gene (gene/Gapdh) was normalized to that of the control group.

### 4.7. Statistical Analysis

All statistical analyses were performed using JMP statistical software, version 14.3 (SAS Institute Inc., Cary, NC, USA). Statistical analyses were performed using a two-way ANOVA and Dunnett’s test. Values of *p* < 0.05 were considered to be statistically significant. Results are expressed as means ± standard deviation.

## 5. Conclusions

Administration of BLM + LAP induced the thickening of hepatic peritoneal tissue via the synergistic or additive effect of lansoprazole and bleomycin. This mechanism might be associated with the migration of macrophages and eosinophils via Mcp1 and Mcp3 secretion in fibroblasts treated by both drugs and the accumulation of collagen via the fibroblasts activated by both drugs. In addition, the slight induction of the MMT by bleomycin might be enhanced by lansoprazole. The peritoneal thickening seen here is different to the animal model for peritoneal sclerosis resulting from peritoneal dialysis, and it could become a novel peritoneal sclerosis model.

## Figures and Tables

**Figure 1 ijms-24-16108-f001:**
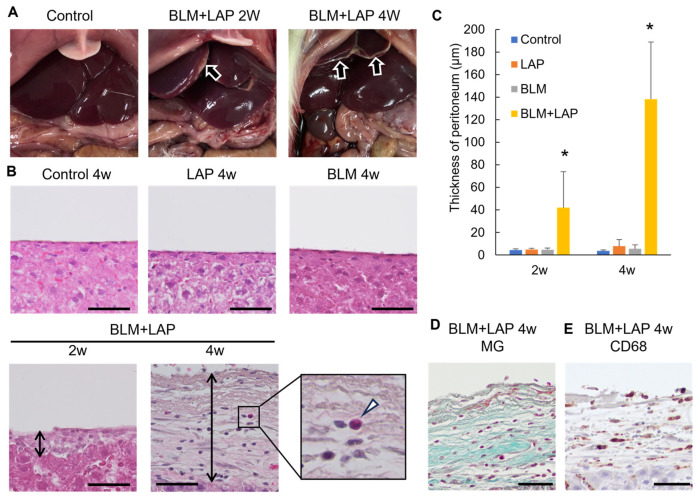
Peritoneal thickening induced by administration of BLM + LAP. We observed peritoneal thickening at the edge of the liver. The phenotype is indicated by arrows (**A**). Peritoneal thickening was induced only by administration of BLM + LAP. Double arrows indicate the thickness of peritoneum. Arrowhead indicates eosinophils in the sub-mesothelial connective tissue of the liver thickened by administration of BLM + LAP (**B**). This increased in a time-dependent manner (**C**). Hematoxylin and eosin stains indicated eosinophils in the thickened peritoneal tissue (**B**). Masson–Goldner stains indicated collagen fibers in the thickened peritoneal tissue (**D**). Immunohistochemistry using the anti-CD68 antibody indicated macrophages in the thickened peritoneal tissue (**E**). Scale bars indicate 50 μm. Asterisks indicate significant difference compared with the control (*p* < 0.05). Five rats each were allocated to the control, lansoprazole, and bleomycin groups, and six rats were allocated to the BLM + LAP group. *p* values were calculated by Dunnet’s test. LAP: lansoprazole, BLM: bleomycin.

**Figure 2 ijms-24-16108-f002:**
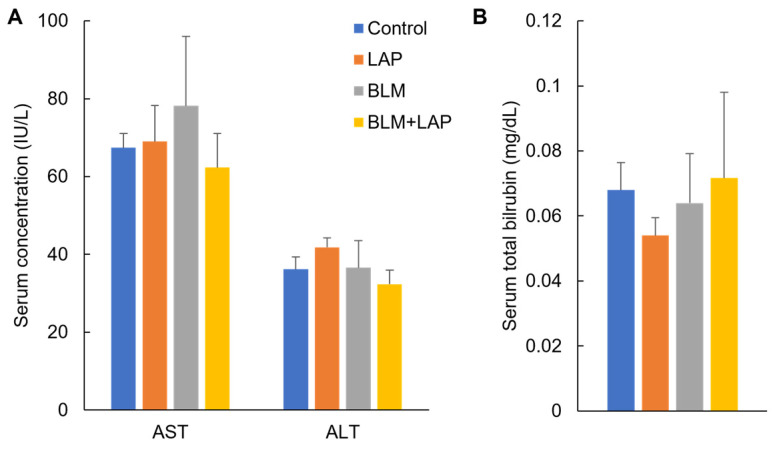
Administration of BLM + LAP for four weeks did not induce liver injury or inhibit the secretion of bile. The concentrations of ASL and ALT, markers of damaged hepatic cells, were unchanged in serum (**A**), and there was no change in the concentration of total bilirubin, the marker of bile stasis, in serum (**B**). Five rats each were allocated to the control, lansoprazole, and bleomycin groups, and six rats were allocated to the BLM + LAP group. *p* values were calculated by Dunnet’s test.

**Figure 3 ijms-24-16108-f003:**
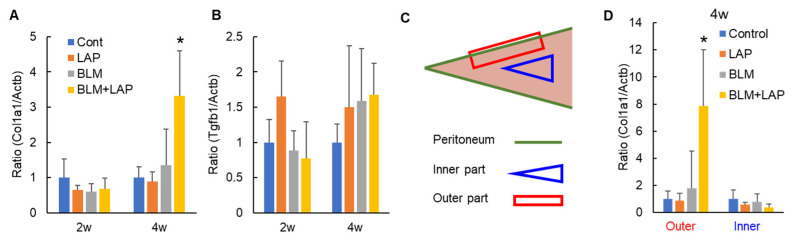
Expression changes of genes associated with fibrosis in the livers of rats given BLM + LAP. The expression of Col1a1 changed in a time-dependent manner (**A**). The expression of Tgfb1 was not changed by bleomycin or lansoprazole separately (**B**). To examine whether the peritoneal thickening was associated with Col1a1 or Tgfb1 in detail, the liver samples were divided into two groups: samples from the inner parts and samples from the outer parts (**C**). Col1a1 was only upregulated by BLM + LAP in the outer part of liver (**D**). Five rats each were allocated to the control, lansoprazole, or bleomycin groups, and six rats were allocated to the BLM + LAP group. One rat in the BLM + LAP group was excluded due to a missing sample. *p* values were calculated by Dunnet’s test. Asterisks indicate a significant difference from the control group (*p* < 0.05).

**Figure 4 ijms-24-16108-f004:**
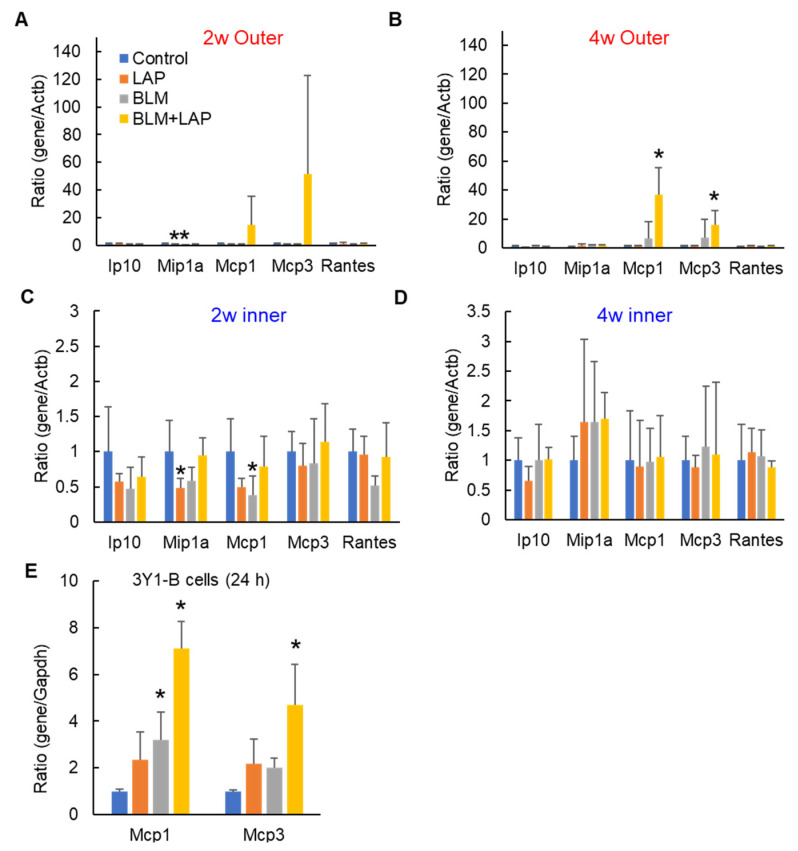
Expressions of chemokines inducing the migration of macrophages or eosinophils in the liver. The expression of chemokines in fibroblast cells was measured in the outer (**A**,**B**) and inner parts (**C**,**D**). Expressions of *Mcp1* and *Mcp3* were measured in 3Y1-B cells (**E**). These chemokines promote the migration of macrophages, and Mcp3 and Rantes also promote the migration of eosinophils. Asterisks indicate a significant difference from the control group (*p* < 0.05).

**Figure 5 ijms-24-16108-f005:**
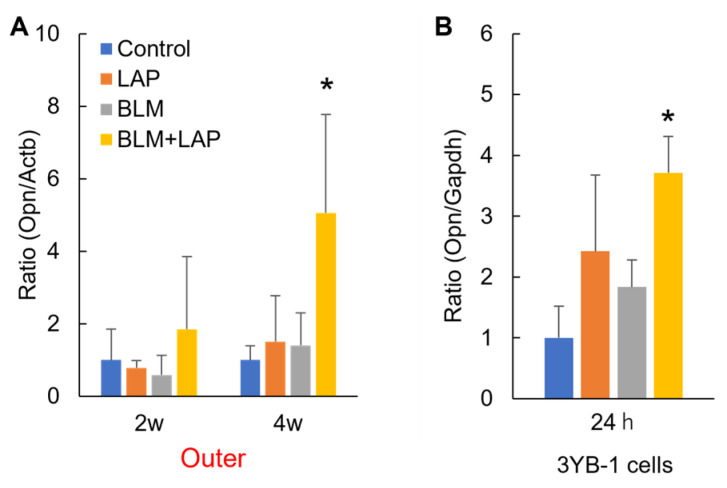
BLM + LAP increased the expression of the *Opn* gene. The expression of the *Opn* gene in activated fibroblasts was increased by BLM + LAP in the outer part of liver (**A**) and in 3Y1-B cells (**B**). Five rats each were allocated to control, lansoprazole, and bleomycin groups, and six rats were allocated to the BLM + LAP group. Cell culture experiments were performed four times. *p* values were calculated by Dunnet’s test. Asterisks indicate a significant difference from the control group (*p* < 0.05).

**Figure 6 ijms-24-16108-f006:**
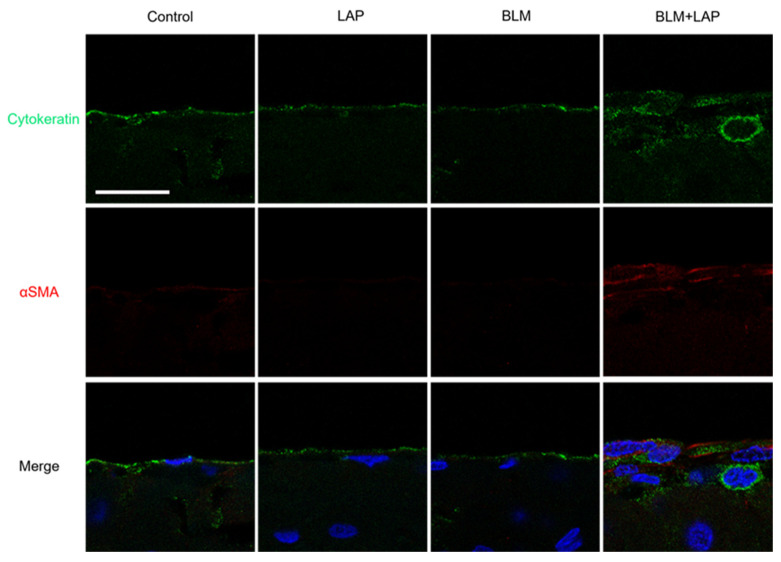
Administration of BLM + LAP was associated with mesothelial–mesenchymal transition. The peritoneal tissue around the liver was double-stained for cytokeratin (green) and αSMA (red) followed by DAPI (blue). Scale bar indicates 60 μm.

**Figure 7 ijms-24-16108-f007:**
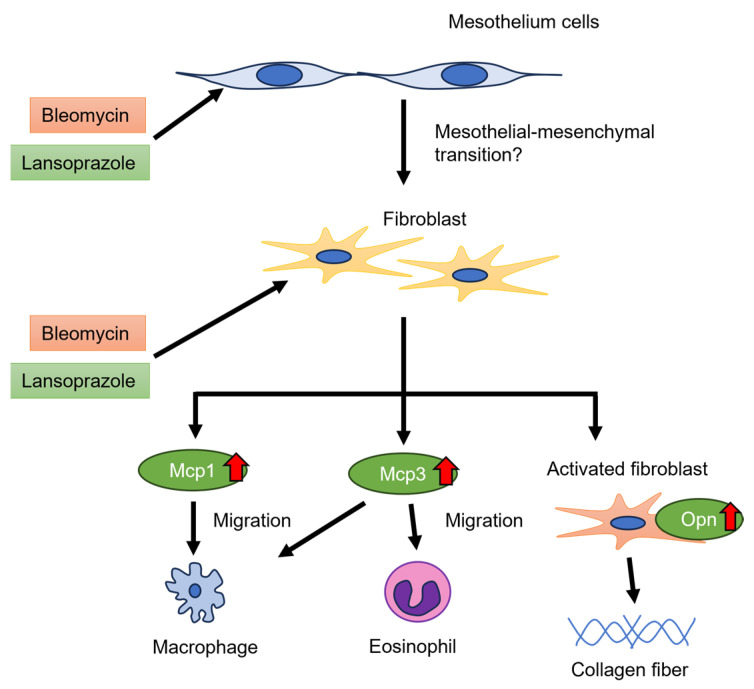
Scheme of this study.

**Table 1 ijms-24-16108-t001:** Primer sequences for quantitative PCR.

Symbol	Forward	Reverse
Col1a1	AGGCTGGTGTGATGGGATT	AGGGCCTTGTTCACCTCTCT
Tgfb1	GACCGCAACAACGCAAT	GGCACTGCTTCCCGAAT
HO-1	ACAGGGTGACAGAAGAGGCTAA	CTGTGAGGGACTCTGGTCTTTG
Cat	GCCTGTGTGAGAACATTGC	CCTGTACGTAGGTGTGAATTG
Gsta2	CTTCTCCTCTATGTTGAAGAGTTTG	TTTTGCATCCACGGGAA
Nqo1	CAGCGGCTCCATGTACT	GACCTGGAAGCCACAGAAG
Gpx1	GGACTACACCGAAATGAATGAT	CTCGCACTTCTCAAACAATG
Mcp1	TTG TCA CCA AGC TCA AGA GA	CAC ATT CAA AGG TGC TGA AG
Mcp3	GCATGGAAGTCTGTGCTGAA	CGTTCCTACCCCTTAGGAC
Ip-10	TCCTGCAAGTCTATCCTGTC	TGGCTTCTCTCTAGTTACGG
Mip-1a	GCGAGTACCAGTCCCTTCTC	GGTGCTGAGCAGGTAACAGA
Rantes	TCGTCTTTGTCACTCGAAGG	GAGCAAGCAATGACAGGAAA
Opn	AGTGGTTTGCCTTTGCCTGTT	TCAGCCAAGTGGCTACAGCAT
Actb	GGAGATTACTGCCCTGGCTCCTA	GACTCATCGTACTCCTGCTTGCTG
Gapdh	AGGTTGTCTCCTGTGACTTC	CTGTTGCTGTAGCCATATTC

## Data Availability

Data are contained within the article and Appendix A.

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
