# Peer review of "Novel Peritoneal Sclerosis Rat Model Developed by Administration of Bleomycin and Lansoprazole"

_ijms, 2023, doi:10.3390/ijms242216108_

Round 1
Reviewer 1 Report
Comments and Suggestions for Authors
This study demonstrated that administration of BLM+LAP for up to 4 weeks peritoneal thickening of tissue around rat livers. This occurs as a result of the infiltration of macrophage and eosinophils and the accumulation of collagen fiber. This study is relevant to the thickness cause by peritoneal dialysis in a time dependent manner. However, the study offers little explantion how this can be prevented in the case of peritoneal dialysis. There are some minor comments to address
Comments:
1. Line 55 with is missing a word “genes associated with the histological changes”
2. What is panel A and B showing in Fig 2.
3. Where is panel E in Figure 3?
4. Bleomycin is known to induce ROS—is the peritoneal thickening of the liver a result of ROS and would antioxidant such as ascorbic acid block the effect?
5. In addition, bleomycin can induced DNA damage single strand and double strand breaks. Would the peritoneal thickening occur if another DNA damaging agent is used such as cisplatin?
6. The discussion is excessively long with many parts repeating the results—it would be more informative to provide a bit more explanation of the data and possible ways to prevent peritoneal thickening caused by dialysis.
Author Response
Reviewer 1
This study demonstrated that administration of BLM+LAP for up to 4 weeks peritoneal thickening of tissue around rat livers. This occurs as a result of the infiltration of macrophage and eosinophils and the accumulation of collagen fiber. This study is relevant to the thickness cause by peritoneal dialysis in a time dependent manner. However, the study offers little explantion how this can be prevented in the case of peritoneal dialysis. There are some minor comments to address
Comments:
- Line 55 with is missing a word “genes associated with the histological changes”
Thank you for your careful review and comment. We have added “with” to the revised manuscript.
- What is panel A and B showing in Fig 2.
We apologize for the omission in the original version. Explanation has been added to the legend. The added explanations is below.
“The concentration of ASL and ALT, markers of damaged hepatic cells, was unchanged in serum (A), and there was no change in the concentration of total bilirubin, the marker of bile stasis, in serum (B). Five rats each were allocated to the control, lansoprazole, and bleomycin groups, and six rats were allocated to the BLM+LAP group. P values were calculated by Dunnet’s test.”
- Where is panel E in Figure 3?
We apologize for the superfluous text that appeared in the legend of Figure 3. The explanation of ‘Fig. 3E’ has been removed from the revised manuscript. The panels were intended to be A-D.
- Bleomycin is known to induce ROS—is the peritoneal thickening of the liver a result of ROS and would antioxidant such as ascorbic acid block the effect?
We did not originally indicate that the expression changes of anti-oxidative stress protein genes. This data has been added to the Results section (2.1.2) and to Supplemental Figure 1 of the revised manuscript. Lansoprazole increased the expression of Nqo1, one of anti-oxidative stress protein, gene, and the expression was also increased by bleomycin, because ROS induced by bleomycin might increase the expression of Nqo1 gene. Serum AST and ALT levels were, however, also unchanged in any groups compared with the control group. Administration of either or both of the drugs is not thought to induce cellular damage in the liver. Bleomycin is an agonist of TLR2 as well as an inducer of ROS and it increases the expression of genes associated with fibrosis and proinflammatory cytokines (Yang et. al., J Immunol 2009, 182, 692-702). We therefore believe that the peritoneal sclerosis might be more strongly associated with the agonist of TLR2 than the inducer of ROS. Nevertheless, it still remains that high dose of ascorbic acid as an antioxidative agent might indeed have ameliorated the peritoneal sclerosis. This concern will be addressed in a future study.
- In addition, bleomycin can induced DNA damage single strand and double strand breaks. Would the peritoneal thickening occur if another DNA damaging agent is used such as cisplatin?
Thank you for the interesting comments. We have not examined whether cisplatin and lansoprazole induced the peritoneal sclerosis. A previous review reported that intraperitoneal administration of cisplatin induced the sclerosis encapsulating peritonitis in several case reports (Ni et al., see below). This study demonstrated that the administration of bleomycin and lansoprazole subcutaneously induced the peritoneal sclerosis. Cisplatin is an inducer of ROS but not an agonist of TLR2. We therefore suggest that the administration of cisplatin and lansoprazole subcutaneously do not induce the peritoneal sclerosis, but confirmation in future studies is necessary.
Ni Z, Chen Q, Huang C, Wang S, Huang Q. Sclerosing encapsulating peritonitis as a rare cause of intestinal obstruction after the treatment of peritoneal mesothelioma: a case report and review of the literature. Transl Cancer Res. 2021 Jun;10(6):3074-3080. doi: 10.21037/tcr-20-3259. PMID: 35116616; PMCID: PMC8798027.
- The discussion is excessively long with many parts repeating the results—it would be more informative to provide a bit more explanation of the data and possible ways to prevent peritoneal thickening caused by dialysis.
Thank you for your advice. In response to your helpful comment, we have carefully revised the discussion section in the revised manuscript and removed a lot of the repetition.

Reviewer 2 Report
Comments and Suggestions for Authors
Kunitatsu et al. present a prospective, in vivo study involving the administration of bleomycin and lansoprazole to rats to cause peritoneal sclerosis. The rationale for the investigation was the serendipitous finding that using lansoprazole, an anti-inflammatory agent, to prevent the inflammation caused by bleomycin in rats had the unanticipated effect of causing thickening of the peritoneum. This would, of course, be an unwanted side effect. Thus, the authors set out to define the time course and potential mechanism of this phenomena.
Rats were administered vehicle, either medication separately, or both medications together for two or 4 weeks. Blood based assays of end organ injury and gene expression were obtained, gross anatomical findings were documented, and histological assessments of the peritoneum with immunohistochemistry was performed. Relevant cell culture experiments with 3Y1-B cells were performed.
The data presented seem to demonstrate that only the combination of medicines, in a time-dependent manner, caused the peritoneum covering the liver to thicken secondary to collagen deposition, with increased eosinophils present. The thickening appeared to primarily affect the outer layer of peritoneum. The upregulation of posited genes occurred in both peritoneal cells and cell culture when exposed to both medicines. No important changes in markers of injury (ALT, AST) were observed between the groups. The immunohistochemistry findings are consistent with transformation of mesenchymal cells into fibroblasts as illustrated in figures 6 and 7.
Based on changes in the aforementioned markers, the authors posit that this model is similar to that of peritoneal dialysis mediated peritoneal thickening, albeit via very different agents causing similar gene expression.
Overall, this is an interesting manuscript. However, there are a variety of issues to be addressed.
Results
1. Figures. Please provide the statistics used and number of animals or replicates of cell culture experiments used in the figure legends.
Methods
1. The use of unpaired t-test is inappropriate when more than two conditions are involved. I do not see any data sets presented that should be assessed with this test. The authors must use one-way analysis of variance (ANOVA) and then use the Dunnett’s post hoc if they wish, although using a Tukey equivalent would be better.
2. Given the small number of animals and replicates, and the variance observed, what was the post hoc statistical power? Should the authors have used more animals to have a power > or = to 0.8 for P<0.05.
In summary, while promising, until the statistical issues are resolved the conclusions cannot be adequately supported.
Author Response
Reviewer 2
Kunitatsu et al. present a prospective, in vivo study involving the administration of bleomycin and lansoprazole to rats to cause peritoneal sclerosis. The rationale for the investigation was the serendipitous finding that using lansoprazole, an anti-inflammatory agent, to prevent the inflammation caused by bleomycin in rats had the unanticipated effect of causing thickening of the peritoneum. This would, of course, be an unwanted side effect. Thus, the authors set out to define the time course and potential mechanism of this phenomena.
Rats were administered vehicle, either medication separately, or both medications together for two or 4 weeks. Blood based assays of end organ injury and gene expression were obtained, gross anatomical findings were documented, and histological assessments of the peritoneum with immunohistochemistry was performed. Relevant cell culture experiments with 3Y1-B cells were performed.
The data presented seem to demonstrate that only the combination of medicines, in a time-dependent manner, caused the peritoneum covering the liver to thicken secondary to collagen deposition, with increased eosinophils present. The thickening appeared to primarily affect the outer layer of peritoneum. The upregulation of posited genes occurred in both peritoneal cells and cell culture when exposed to both medicines. No important changes in markers of injury (ALT, AST) were observed between the groups. The immunohistochemistry findings are consistent with transformation of mesenchymal cells into fibroblasts as illustrated in figures 6 and 7.
Based on changes in the aforementioned markers, the authors posit that this model is similar to that of peritoneal dialysis mediated peritoneal thickening, albeit via very different agents causing similar gene expression.
Overall, this is an interesting manuscript. However, there are a variety of issues to be addressed.
Results
- Figures. Please provide the statistics used and number of animals or replicates of cell culture experiments used in the figure legends.
Thank you for your careful analysis and helpful comments. The numbers of animals in animal studies and the number of replicates in cell culture experiments have been added to the legends and to sections 4.3 and 4.4, respectively.
Methods
- The use of unpaired t-test is inappropriate when more than two conditions are involved. I do not see any data sets presented that should be assessed with this test. The authors must use one-way analysis of variance (ANOVA) and then use the Dunnett’s post hoc if they wish, although using a Tukey equivalent would be better.
All statistical analyses were performed using Dunnett's test because only the BLM+LAP group were the remarkable thickness of peritoneum. To examine how BLM+LAP induced peritoneal sclerosis, we measured the expression of genes associated with the phenotype in BLM+LAP group to compare control group. Those in the lansoprazole group or in the bleomycin group were measured to examine whether either drug induced the expression changes of these genes as well as BLM+LAP group. The aim of this study is not the comparison of peritoneal sclerosis between all groups, but rather to explore the mechanism of peritoneal sclerosis only in BLM+LAP group.. We therefore performed Dunnet’s test but not Tukey HSD test.
Some multiples including Scheffe’s test comparisons were performed following ANOVA because they required F value calculated from ANOVA. Dunnet’s test and Tukey HSD test do not require ANOVA, because F values are not needed to calculate the p value in these multiple comparisons. Therefore, ANOVA was not performed in this study.
The sentences about performing t test have been removed from the revised manuscript.
- Given the small number of animals and replicates, and the variance observed, what was the post hoc statistical power? Should the authors have used more animals to have a power > or = to 0.8 for P<0.05.
This study used 42 rats to perform two kinds of time course. Five or six rats were allocated in each group. Past studies that prepared peritoneal sclerosis model rats allocated five rats in each group (Komatsu et al. 2008; Liu et al. 2022). If animals were small, significant difference might occasionally be masked. However, peritoneal thickness, expression of Col1a1, Mcp1, Mcp3 and Opn were significantly increased in BLM+LAP group. As Reviewer 2 pointed out, variance of expression changes was observed in Tgfb1 expression in animal studies (Fig 3B). This concern has therefore been added as a study limitation (Page 9, lines 283-288). Nevertheless, we do not believe the concern directly affects the conclusion of our study.
- Komatsu H, Uchiyama K, Tsuchida M, Isoyama N, Matsumura M, Hara T, Fukuda M, Kanaoka Y, Naito K. Development of a peritoneal sclerosis rat model using a continuous-infusion pump. Perit Dial Int. 2008 Nov-Dec;28(6):641-7. PMID: 18981396.
- Liu Y, Ma Z, Huang Z, Zou D, Li J, Feng P. MiR-122-5p promotes peritoneal fibrosis in a rat model of peritoneal dialysis by targeting Smad5 to activate Wnt/β-catenin pathway. Ren Fail. 2022 Dec;44(1):191-203. doi: 10.1080/0886022X.2022.2030360. PMID: 35170385; PMCID: PMC8856067.
In summary, while promising, until the statistical issues are resolved the conclusions cannot be adequately supported.
In this study, we concluded that administration of BLM+LAP induced the thickness of hepatic peritoneal tissue. The accumulation of collagen fiber and the migration of macrophages and eosinophils were observed in thickened peritoneum as well as the encapsulating peritoneal sclerosis. The expression of phenotype associated genes (Col1a1, Mcp1, Mcp3 and Opn) was significantly increased. Thus, the statistical concerns indicated by Reviewer 2 may not have directly affected the conclusion. Nonetheless, the concerns pointed out by Reviewer 2 have been added as a study limitation in Discussion section of the revised manuscript.

Round 2
Reviewer 2 Report
Comments and Suggestions for Authors
4.7 Statistical analysis
All statistical analyses were performed using JMP version 14.3 statistical software (SAS Institute Inc., Cary, NC, USA). Statistical analyses were performed using Dunnett’s test. Values of P <0.05 were considered to be statistically significant. Results are expressed as mean ± standard deviation.
This section is copied from the author's manuscript.
Dunnett's test is one of a number of a posteriori or post hoc tests, run after a significant one-way analysis of variance (ANOVA), to determine which differences are significant. The authors cannot use t-tests as there is more than two conditions present - there are 4. They have to use ANOVA. Further, as 4 variates are involved (no medicine, one or the other medicine, two medicines), the authors must use two-way ANOVA.
Like it or not, the design is classic to prove or disprove linkage/synergy. The authors have shown synergy. Two medicines make the difference, either one does not. This is a classic model for two-way ANOVA. The authors can prove synergy and interaction with two-way ANOVA, something that t-tests cannot prove. Please use the correct statistics. The authors are directed to seek statistical assistance.
Author Response
Reviewer 2
Dunnett's test is one of a number of a posteriori or post hoc tests, run after a significant one-way analysis of variance (ANOVA), to determine which differences are significant. The authors cannot use t-tests as there is more than two conditions present - there are 4. They have to use ANOVA. Further, as 4 variates are involved (no medicine, one or the other medicine, two medicines), the authors must use two-way ANOVA.
Like it or not, the design is classic to prove or disprove linkage/synergy. The authors have shown synergy. Two medicines make the difference, either one does not. This is a classic model for two-way ANOVA. The authors can prove synergy and interaction with two-way ANOVA, something that t-tests cannot prove. Please use the correct statistics. The authors are directed to seek statistical assistance.
Thank you for your comments. We reconsulted with a biological statistician, and he said:
“Either two-way ANOVA or Dunnett’s test was needed in this study. But two-way ANOVA can indicate which the phenotype might be induced by the synergy or additive effect.”
This advice is consistent with your advice, and we performed two-way ANOVA. In the revised manuscript we have added the results of two-way ANOVA and modified the Discussion and Conclusion sections. The additional and modified parts have been marked in red. The revised manuscript has been carefully polished up and we think it has been greatly improved by your helpful comments. Thank you very much.
Round 3
Reviewer 2 Report
Comments and Suggestions for Authors
Thank you for doing what I asked. Your results are fantastic visually, and the statistics support these results as expected. Well done.